# Effect of Physical Form and Level of Wheat Straw Inclusion on Growth Performance and Blood Metabolites of Fattening Goat

**DOI:** 10.3390/ani10101861

**Published:** 2020-10-13

**Authors:** Muhammad I. Malik, Muhammad A. Rashid, Muhammad S. Yousaf, Saima Naveed, Khalid Javed, Habib Rehman

**Affiliations:** 1Department of Animal Nutrition, University of Veterinary and Animal Sciences, Lahore 54000, Pakistan; dr.irfan279@gmail.com (M.I.M.); saimamahad@uvas.edu.pk (S.N.); 2Department of Physiology, University of Veterinary and Animal Sciences, Lahore 54000, Pakistan; drmshahbaz@uvas.edu.pk (M.S.Y.); habibrehman@uvas.edu.pk (H.R.); 3Department of Livestock Production, University of Veterinary and Animal Sciences, Lahore 54000, Pakistan; khalidjaved@uvas.edu.pk

**Keywords:** conventional, high concentrate, nitrogen balancing, pelleted, total mixed ration

## Abstract

**Simple Summary:**

The growth of small ruminant production is critical as the increasing human population continues to consume meat. An intensive and sustainable fattening system requires faster gains and fewer labor requirements at a lower cost. The current experiment was planned to evaluate the effects of pelleted and conventional total mix ration (TMR) with 15% and 25% wheat straw on growth performance, nutrient digestibility, nitrogen balancing, liver enzymes, blood metabolites, and complete blood counts. The feed intake and growth performance improved in both pelleted TMR treatments. The rumen pH was comparatively higher in those goats fed conventional TMR; however, in pelleted TMR the pH was greater than the threshold level set for subacute ruminal acidosis. Nutrient digestibility, nitrogen balancing, blood metabolites, complete blood count, and liver enzymes were not influenced by straw level or physical form of diet.

**Abstract:**

The inclusion of straw in high concentrate total mixed rations (TMRs) of male fattening goats can provide the necessary fiber to prevent ruminal acidosis and maintain growth. The objective of this study was to evaluate the effects of the physical form (PF) of the diet (pelleted vs. conventional) and the straw level (SL) of wheat straw (WS) (15% versus 25%) in total mixed rations on feed intake, growth, total tract digestibility, and blood metabolites of fattening goats. Thirty-two male Beetal goats (27.4 ± 0.28 kg body weight (BW)) were divided randomly into the following four dietary treatments with a 2 × 2 factorial arrangement (*n* = 8/treatment): (1) CTMR15 (conventional TMR containing 15% WS), (2) CTMR25 (conventional TMR containing 25% WS), (3) PTMR15 (pelleted TMR containing 15% WS), and (4) PTMR25 (pelleted TMR containing 25% WS). Both conventional and pelleted 15% WS TMR had 33.7% neutral detergent fiber (NDF) and 19.3% acid detergent fiber (ADF), whereas in 25% WS TMR the NDF and ADF contents were 38.7% and 22.9%, respectively. The experimental diets were formulated to be iso-nitrogenous (crude protein (CP) = 15%). The dry matter intake (DMI) (1.265 vs. 1.044 kg/day) and average daily gain (ADG) (0.176 vs. 0.143 kg/day) were higher (*p* < 0.05) in pelleted vs. conventional TMR-fed goats. Irrespective of the PF of the TMR, the 15% WS-fed animals had greater (*p* < 0.05) DMI (1.206 vs. 1.102 kg/day) and ADG (0.172 vs. 0.144 kg) when compared to those fed on 25% WS diets. Furthermore, feed-to-gain ratio (F:G) was higher (*p* < 0.05) in the 25% WS-fed goats when compared with the 15% WS-fed animals. Digestibility coefficients, nitrogen balancing, hepatic enzymes, blood metabolites, and hematological parameters were similar (*p* > 0.05) across all treatments. In conclusion, feeding pelleted TMR with WS improved DMI and growth performance as compared to those fed conventional TMR, and 15% WS performed better than 25% WS without exerting any adverse effects on blood metabolites, liver enzymes, or hematological parameters.

## 1. Introduction

Fattening animals are fed on high concentrate diets to achieve faster growth rates and to reduce the duration of the fattening period [1]. Total mixed ration (TMR) is a commonly used method of feeding livestock that ensures a balanced consumption of nutrients, minimizes the chances of feed selection, and improves animal performance by stabilizing the ruminal environment [2]. In developing countries, crop residues—mainly wheat straw (WS)—constitute a key component of livestock feeding [3]. Previously, in sheep and goats, WS inclusion was evaluated up to 15% [4], 20% and 40% [5], 60% and ad libitum [6], the increase in forage-to-concentrate ratio, decreases growth performance, and increase in cost of production [7]. Wheat straw has high neutral detergent fiber (NDF) contents (81%), which can help to stabilize the ruminal environment of fattening animals fed on diets with a higher concentrate level. Recent developments in feeding processing have suggested that straw sources can be ground and successfully pelleted as a TMR. Straw-based pelleted TMR is an innovative technology that minimizes the issues related to handling, storage, and transportation of bulky straws [8,9].

It is well established that forage-to-concentrate ratio and physical form (PF) of the diet can have a significant influence on intake, growth performance, and nutrient utilization. Haddad [7] reported that goat kids fed a diet with a high concentrate level (15:85) improved ADG and lowered the cost of feeding compared to those fed on diets with higher forage levels (60:40, 45:55, 30:70). Blanco et al. [10] reported that pelleted total mixed ration (PTMR) containing 25% ground barley straw had the highest DMI and ADG and the lowest duration of fattening in lambs. In the same study, animals fed on 15% WS-based PTMR experienced a lower DMI, ADG, and digestibility of DM and a drop in rumen pH, which may be due to the fine particle size of straw (2 mm) in pelleted TMRs, suggesting the importance of the particle size of straw in pelleted TMR. A possible way to stabilize the ruminal environment is either by increasing the level or the particle size of straw in the TMR. The particle size and straw concentration in the fattening rations can influence the physical effective NDF (peNDF) contents of the diet, which has a direct impact on chewing activity, buffering capacity, and welfare of the animals [11]. Most of the studies advocating the use of PTMR have been carried out on lambs using straw finely ground at 2 mm [10] and 3 mm [9]. The fine grinding of straw sources at 2 or 3 mm particle size is laborious and time consuming and increases the cost of grinding, which could be a potential limitation to its adoption by the feed manufacturing industry. In Pakistan, the particle size of WS obtained after wheat threshing is approximately 8 mm, which may provide an opportunity for its effective utilization in fattening TMRs. To our knowledge, there is no study comparing the use of WS ground at 8 mm in pelleted and conventional TMR. We hypothesized that, with a relatively larger particle size (8 mm), pelleted TMR with 15% WS (85% concentrate) may improve DMI and growth performance without any negative impact on rumen health and nutrient digestibility. Sheep and goat exhibit different feeding behaviors [12], and the response of fattening goats to PF of the TMR and level of WS is still unclear. Therefore, the current experiment was planned to evaluate the conventional TMR (CTMR) and PTMR containing 15% and 25% WS ground at 8 mm. Other objectives were to determine the effects of PF and WS level on intake, growth performance, nutrient digestibility, and selected blood metabolites. 

## 2. Materials and Methods 

### 2.1. Experimental Design and Animal Husbandry

Experimental procedures were approved by the Animal Care and Use Committee (dr/1214: 09-11–2017), University of Veterinary and Animal Science, Lahore, Pakistan. The experiment was conducted at the Small Ruminant Research and Training Center, UVAS, Ravi Campus, Pattoki, Pakistan. Thirty-two male Beetal goats, weighing 27.4 ± 1.62 kg (mean ± SD), were procured from the market and brought to the research facility. Upon arrival, the animals were treated against ecto- and endoparasites using a sub-cut injection (Dectomax, Pfizer, Brooklyn, NY, USA). Two weeks after deworming, the animals were vaccinated for *Clostridia* (Bar-Vac CD/T, Boehringer Ingelheim, Berlin, Germany) and contagious caprine pleuropneumonia (CCPP, JOVAC, Amman, Jordan) according to farm practices. After initial quarantine procedures, the animals were randomly allotted to four different dietary treatments (*n* = 8 animals/treatment) in a completely randomized design with a 2 × 2 factorial arrangement. One factor was the PF of TMR (conventional vs. pelleted) and another factor was the SL level of WS (15% vs. 25%) in TMR (Table 1). The four dietary treatments were as follows: (1) CTMR15 (conventional TMR containing 15% WS); (2) CTMR25 (conventional TMR containing 25% WS), (3) PTMR15 (pelleted TMR containing 15% WS), and (4) PTMR25 (pelleted TMR containing 25% WS). Wheat straw used in the PTMR was ground to pass through an 8 mm sieve using a hammer mill; concentrate ingredients were ground to 2 mm, and then mixed in a double ribbon horizontal mixer. Mixed material was pelleted at 65 °C with the addition of steam to produce 8 mm × 10 mm (diameter × length) straw-based TMR pellets (). Wheat straw and experimental diets were shaken using a Penn State Particle Separator to obtain four different fractions: long (>19 mm), medium (<19, >8 mm), short (>1.18 mm), and fine (<1.18 mm). The physical effectiveness (pef) factor was calculated as the total proportion of particles retained on three sieves (18 mm, 8 mm, and 1.18 mm) of the Penn State Particle Separator. The peNDF was calculated as dietary NDF content (% DM) multiplied by pef 1.18 [13] (Table 2).

The total experimental duration was 108 days including the first 10 days of dietary adaptability, followed by 91 days for data collection and the last 7 days for digestibility and urine collection. Diets were formulated to be iso-nitrogenous (Table 1) and fed ad libitum. The animals were housed in individual pens (1.5 × 1.4, length × width) and fed twice a day at 06:00 and 18:00. The animals were given free-choice access to fresh and clean water during the entire experiment. 

### 2.2. Feed Intake and Growth Performance

The animals were weighed before morning feeding at the start and then on a weekly basis using a digital weighing balance. Orts were collected daily to determine daily DMI. Body length, heart girth, hip height, and wither height were measured at the start and then on a weekly basis. Body condition scoring was performed at day 0, 30, 60, and 90 of the experiment using a 1–5 scoring system [15].

### 2.3. Fecal Score

The fecal score was conducted on a daily basis using a 1-to-5 scoring system described by Le Jambre et al. [16] with 1 being normal pellets and 5 being watery feces that run on a flat surface and do not maintain a depth.

### 2.4. Digestibility

At the end of the experiment, five goats per treatment were confined individually in digestibility cages to determine the nutrient digestibility. After the initial two days of adaptability to the digestibility cages, individual feed intake, fecal, and urine outputs were recorded daily for five days. Urine samples were collected in a pre-acidified container (50 mL of 5% H_2_SO_4_). A 10% portion of daily fecal and urine outputs was collected in zipper bags and plastic bottles, respectively, and stored at −30 °C until further analysis [10]. A representative sample (500 g) of all experimental diets offered to animals was collected and stored at −30 °C for subsequent nutrient analysis. The frozen fecal and feed refusal samples were thawed overnight at room temperature and then bulked together to create a composite sample for each animal, dried at 55 °C for 72 h in a hot air oven and then ground using a 5 mm sieve and then a 2 mm sieve (Wiley Mill, Arthur H. Thomas, Philadelphia, PA, USA). Urine samples were thawed overnight at room temperature, composited for each animal, and then analyzed for N estimation.

The digestion coefficient for each nutrient was calculated using the following formula:Digestibility (%) = [(dietary intake (of the nutrient) [g/d] − fecal output [g/d])/dietary intake [g/d]] × 100.(1)

The nitrogen balance was calculated as follows:N balance (g/d) = [total N intake (g/d) − (total fecal N output [g/d] + total urinary N output [g/d])].(2)

### 2.5. Chemical Analysis

Samples of experimental diets, refusal, and fecal samples were analyzed for DM [17] (method no. 967.03), ash content [17] (method no. 942.05), and ether extract (EE) (Ankom^®^ TX15, ANKOM Technology, Macedon, NY, USA). The NDF and acid detergent fiber (ADF) contents were determined according to Van Soest, et al. [18] using a filter bag technique (Ankom^®^ 200 Fiber Analyzer, ANKOM Technology, Macedon, NY, USA). Additionally, a heat-stable alpha-amylase and sodium sulfite were used for NDF analysis. The N contents of the feed, refusal, and feces were determined according to the Dumas method [17] (method no. 990.03) (Rapid N Exceed, Nitrogen Analyzer System GmbH, Hanau, Germany). Nitrogen contents of urine were determined by the Kjeldahl method [17] (method no. 984.13). The crude protein (CP) content of the samples was calculated by multiplying the N% content by the factor 6.25.

### 2.6. Rumen pH

Fortnightly rumen liquor samples were collected at 4 h after morning feeding using an oral tube [19]. To minimize the possible chances of saliva contamination, the first 200 ml portion of the collected rumen fluid was discarded [20]. Collected rumen fluid was filtered using a four-layered cheesecloth and immediately analyzed for pH (Starter 3100, OHAUS, Parsippany, NJ, USA). 

### 2.7. Blood Collection and Analysis

Blood samples were collected weekly from the jugular vein in an EDTA vacutainer 4 h post morning feeding. Blood samples were centrifuged at 3000× *g* at −4 °C for 15 min. Harvested plasma was preserved in duplicate microfuge tubes and stored at −20 °C until further analysis. Plasma samples were analyzed for glucose (GLUCOSE, 23503 © Biosystems, Barcelona, Spain), blood urea nitrogen (BUN) (BUN, 21516^©^ Biosystems, Barcelona, Spain), and cholesterol (12505 CHOLESTEROL© Biosystems, Barcelona, Spain) using colorimetric kits with the help of a spectrophotometer (Epoch2, BioTek, Winooski, VT, USA). Additionally, at days 30, 60, and 90 of the experiment, blood samples were collected in plain and EDTA vacutainers for liver function test and complete blood count (CBC), respectively. Serum was obtained and stored at −20 °C from blood samples collected in plain vacutainers and centrifuged at 3000× *g* at −4 °C for 15 min. The serum samples were analyzed by an automatic chemistry analyzer (Altair™ 240, Labcompare, South San Francisco, CA, USA) to determine alkaline phosphatase (ALP), alanine transaminase (ALT), aspartate transaminase (AST), and bilirubin. Blood samples collected in EDTA vacutainers were analyzed for CBC, red blood cells (RBCs), white blood cells (WBCs), hemoglobin, lymphocytes, monocytes, mean corpuscle volume (MCV), mean corpuscular hemoglobin concentration (MCHC), and hematocrit (Hct) by using an automatic hematology analyzer (HT-300 3-Diff Auto Hematology Analyzer, MR International Healthcare Technology, Hong Kong). 

### 2.8. Statistical Analysis

All the data were first tested for normality using QQ plots (SAS v. 9.4, University Edition, SAS Institute Inc., Cary, NC, USA). Data for growth performance, digestibility coefficients, nitrogen balancing blood metabolites, serum enzymes, and CBC were analyzed using MIXED Procedures of SAS (SAS v. 9.4, University Edition SAS Institute Inc., Cary, NC, USA). The individual goats were considered as an experimental unit. The model included the following: fixed effects of physical form (PF), straw level (SL), and interaction of PF×SL. For multiple observations between weeks, based on the Akaike information criterion values, the autoregressive type 1 procedure within repeated measure was used for ADG, DMI, rumen pH, fecal score and blood metabolites, liver function test, and hematological parameters. Significance was declared at *p* < 0.05, and LS means were compared by Tukey’s test.

## 3. Results

### 3.1. Growth Performance 

Daily DMI and total DMI were affected by the PF of diet and SL. Goats fed on the PTMR had a greater DMI (*p* < 0.05) as compared to that on the CTMR (Table 3). Higher total DMI and daily DMI were observed for the goats fed 15% WS treatment regardless of the PF (*p* < 0.05) of diet. Initial body weight (BW) was similar (*p* > 0.05) in all treatments. Final BW, the total gain in BW, and ADG were higher (*p* < 0.05) in goats fed the PTMR and the 15% WS TMR than the CTMR and 25% WS TMR -fed goats, respectively. Data for the repeated measure are not shown in the tables; only DMI and ADG were significant for treatment × week interaction. The feed-to-gain ratio (F:G) was lower (*p* < 0.05) in the 15% WS as compared to the 25% WS-fed goats. Fecal consistency was influenced by SL with a higher fecal score (*p* = 0.04) in the 15% WS than that of the 25% WS treatment. Rumen pH was higher (*p* = 0.053) in the CTMR as compared to the PTMR. However, gain in body condition score (BCS) and structural measurements were not affected (*p* > 0.05) by the PF or the SL of the diet (Table 4).

### 3.2. Digestibility Coefficients and Nitrogen Balancing

Nutrient digestibility coefficients, namely dry matter (DM), organic matter (OM), CP, NDF, ADF, and EE (Table 5), were similar (*p* > 0.05) in all treatments. Nitrogen balance, nitrogen intake, fecal nitrogen, urinary nitrogen, and retained nitrogen (Table 6) were also not influenced (*p* > 0.05) by the PF or the SL in the diet.

### 3.3. Liver Enzymes, Blood Metabolites, and Hematological Parameters

Activities of the serum enzymes ALP, ALT, AST and of bilirubin (Table 7) were similar (*p* > 0.05) across all the treatments regardless of the PF of diet and SL of WS. Similarly, blood glucose, BUN, and cholesterol were not different (*p* > 0.05) among treatments. Furthermore, the red blood cells, WBCs, hemoglobin, lymphocytes, monocytes, MCV, MCHC, and hematocrit (Hct) were also similar and within the range of reference values (Table 8).

## 4. Discussion

To avoid rumen acidosis, diets were carefully designed to achieve relatively high NDF contents by adding the WS and soyhulls in all diets (Table 1). Therefore, the NDF contents of 15% WS-based TMRs were 34%, which was greater than the NDF contents (32%) of 25% barley straw-based TMR used by Blanco et al. [10]. To ensure a consistent and uniform supply of nutrients, a high pellet durability index was achieved (90% and 89% in PTMR15 and PTMR25 diets, respectively) due to 10% wheat inclusion [22].

As expected, the DMI was 21.2% higher in PTMR as compared to goats fed on CTMR. A higher DMI of pelleted diet is in line with past studies [8,9]. This higher DMI of PTMR can be attributed to the smaller particle size of ground hay leading to a greater passage rate, a lesser gut filling effect, and a delay in satiety signal [8]. The DMI was also influenced by the SL of the diet, and it was 9.43% higher in goats fed 15% than 25% WS TMR rations. Dry matter intake is highly dependent on the NDF contents of the diet [23]. Irrespective of the PF of the diet, a higher DMI in 15% WS TMR treatment might be due to lower NDF contents in 15% WS diet as compared to 25% WS diets (34 vs. 38, Table 1). The ADG of goats was 23% higher when they were fed pelleted as compared to conventional TMR rations mainly due to greater DMI. Similar observations were documented by Zhang et al. [9] in lambs. Irrespective of the PF of diet, goats fed 15% WS TMRs had 21% greater ADG than the 25% WS TMR-fed goats. Similar to our findings, an increase in forage from 15% to 30% lowered the ADG in kids fed alfalfa hay-based TMR [7]. The feed-to-gain ratio was influenced by the SL of the diet, and a lower feed-to-gain ratio was observed in 15% WS than in 25% WS. An improved feed-to-gain ratio can be attributed to a lower forage-to-concentrate ratio (15:85 vs. 25:75) in 15% WS rations [7], with a greater intake of digestible nutrients from a high concentrate intake [10]. Digestibility of DM, OM, CP, NDF, ADF, and EE was not affected by dietary treatments. Similar findings were documented previously for OM [24], DM, NDF [8], and EE digestibility [25]. Similarly, our results of NDF and ADF digestibility are in line with Kumari et al. [26]. Nitrogen intake, fecal N, urinary N, and retention of N were similar in all treatments; our results are in line with Blanco et al. [10]. 

Rumen pH was slightly lower in PTMR vs. CTMR rations (6.43 vs. 6.49), but it was still not in the range of subacute ruminal acidosis (SARA) (<5.6). Comparatively higher pH in pelleted diets even at 85% concentrate (15% WS TMRs) is associated with a longer (8 mm) particle size of straw as compared to previous experiments [8], suggesting the impact of WS particle size on stabilizing the rumen environment. Unlike cattle and sheep, goats can stabilize their rumen pH by modifying their eating and ruminating behavior [27]. It is well established that peNDF contents of a diet regulate the rumen pH by increasing the chewing time and saliva production [28]. Llonch, et al. [29] documented that a peNDF level of 6.4% is enough to stimulate chewing activities for the prevention of SARA in fattening steers, which is in line with the findings of this experiment that the highest intake, ADG, and FE were obtained in 15% WS-based TMR with 6.4% peNDF contents. In a work on dairy goats, Jang, et al. [30] evaluated TMRs having 23.85%, 21.71%, and 16.22% peNDF contents and documented no difference in intake, ADG, and chewing activity. Physical form and particle size used in the diet are known to influence rumination behavior and the rumen health of animals, which is a limitation in this study. However, we observed that PF or WS level did not influence rumen pH and non-invasive indicators of SARA (ALP, ALT, AST, blood glucose, BUN, and cholesterol), which supports our findings that animals did not encounter SARA in 15% WS TMR with 6.4% peNDF contents.

The fecal score was higher for goats fed 15% vs. 25% WS (1.17 vs. 1.11) TMRs. The mean fecal consistency score was in a normal range (<1.2), indicating that the animals did not experience SARA. Loose fecal consistency is frequently observed in SARA. During SARA, translocation of *Fusobacterium necrophorum* and *Arcanobacterium pyogenes* from rumen results in hepatic abscesses [31] and increased liver enzyme activities [32]. The liver enzyme activities of AST and ALT in serum are excellent indicators of liver function, and a significant increase in serum ALT and AST has been observed in cows subjected to SARA [33]. The increase in AST, ALT, and bilirubin is associated with liver injury or infection [32]. In this experiment, activities of the liver enzymes (ALT, AST, ALP) and of bilirubin were in normal ranges, suggesting that, despite the higher DMI, the animals fed on pelleted TMR had no adverse effects on liver function. Blood glucose, BUN, and cholesterol were also similar across all treatments. Previous research reported that animals exposed to SARA exhibited a decrease in BUN [34], a reduction in cholesterol, and an increase in glucose [35]. Similarly, a higher number of WBCs, MCV, and MCHC were observed in SARA-affected cows [36]. No change in liver enzyme activities, blood metabolites, and hematological parameters indicates that, in our experiment, the goats did not experience negative effects of SARA at any stage even when their intake was greatest in the 15% straw-based PTMR. We can infer that 15% of WS-based TMR can be safely used in fattening goats to achieve greater production without compromising their health.

## 5. Conclusions

The findings of the current experiment revealed that the pelleting of the WS as TMR resulted in greater DMI and growth performance. Furthermore, the DMI, ADG, and feed-to-gain ratio were higher in 15% WS as compared to 25% TMR. In light of these results, it can be concluded that pelleted TMR with 15% WS ground at 8 mm can be successfully used for intensive rearing of goats. Although WS 15% as compared to 25% performed better, it still remains unclear whether it is associated with the NDF contents of diet or straw particle size. Therefore, further studies are warranted to delineate the effect of the particle size of straw and NDF contents of PTMR on the growth performance and feeding behavior of fattening goats.

## Figures and Tables

**Table 1 animals-10-01861-t001:** Inclusion level of ingredients and chemical composition of conventional and pelleted total mixed rations (TMRs) used in the fattening experiment.

Ingredients (% of Dry Matter)	Total Mixed Rations ^1^	
	WS 15%	WS 25%
Corn grain	23.0	16.0
Wheat grain	10.0	10.0
Soy hulls	13.0	9.0
Corn gluten 30%	20.0	20.0
Soybean meal	9.5	10.5
Wheat straw	15.0	25.0
Sugar cane molasses	5.0	5.0
Mineral mixture ^2^	1.0	1.0
Salt	1.0	1.0
Lime	1.5	1.5
Sodium bicarbonate	1.0	1.0
Chemical composition on dry matter basis		
Dry matter (%)	88.10	89.20
Metabolizable energy ^3^ MJ/kg	10.87	10.37
Crude protein (%)	15.00	15.20
Ether extract (%)	2.90	2.70
aNDF ^4^ (%)	33.70	38.70
ADF ^5^ (%)	19.30	22.90
Ash (%)	8.90	10.30
Calcium (%)	1.0	1.0
Phosphorus (%)	0.58	0.57

WS (wheat straw) 15%, TMR (total mixed ration) with 15% wheat straw. WS 25%, TMR with 25% wheat straw. ^1^ Ingredients and nutritional composition of both the pelleted and conventional TMR were similar. ^2^ Each kg of mineral mixture contains the following: calcium: 155 g; cobalt: 10 mg; copper: 600 mg; ferrous: 1000 mg; iodine: 40 mg; magnesium: 55 g; manganese: 2000 mg; phosphorus: 135 g; sodium: 45 g; selenium: 3 mg; zinc: 3000 mg. ^3^ ME = Metabolizable energy was calculated from [14]. ^4^ aNDF: Neutral detergent fiber inclusive of residual ash, ^5^ ADF: Acid detergent fiber inclusive of residual ash.

**Table 2 animals-10-01861-t002:** Particle size distribution (%) of the experimental diets on a Penn State Particle Separator sieve (long >19 mm, medium <19, >8 mm, short >1.18 mm), and fine (<1.18 mm) fed to male fattening goats.

Particle Size	Treatments ^1^				Wheat Straw
	CTMR15	CTMR25	PTMR15	PTMR25	
Long > 19 mm	-	-	-	-	-
Medium < 19, > 8 mm	8.99	16.81	6.35	14.38	67.16
Short > 1.8 mm	12.30	10.26	12.71	10.79	19.67
Fine < 1.18 mm	78.70	72.92	80.92	74.82	13.17
pef factor ^2^	21.29	27.07	19.07	25.18	86.83
peNDF 1.18	7.18	10.48	6.43	9.74	70.33

^1^ Treatments, CTMR15 = conventional TMR containing 15% wheat straw, CTMR25 = conventional TMR containing 25% wheat straw, PTMR15 = pelleted TMR containing 15% wheat straw, PTMR25 = pelleted TMR containing 25% wheat straw. ^2^ pef = physical effectiveness, the pef factor was calculated as a total proportion of particles retained on three sieves (18 mm, 8 mm, and 1.18 mm) of the Penn State Particle Separator; peNDF = physically effective NDF, calculated as dietary NDF content (% DM) multiplied by pef 1.18 [13].

**Table 3 animals-10-01861-t003:** Growth performance, feed-to-gain ratio, rumen pH, fecal score, and structural measurements in fattening goats fed conventional and pelleted total mixed ration.

Parameters	Treatments				SEM ^3^	*p*–Value ^4^		
	PF		SL					
	Conventional	Pellet	WS15	WS25		PF	SL	PF × SL
DMI (kg/day) ^1^	1.044	1.265	1.207	1.102	0.009	<0.001	<0.001	0.893
Total DMI (kg)	95.00	115.1	109.8	100.3	1.921	<0.001	<0.001	0.924
Body weight (kg)								
Initial	27.36	27.51	27.78	27.10	0.590	0.800	0.255	0.878
Final	40.35	43.54	43.71	40.18	0.623	<0.001	<0.001	0.988
Gain	13.99	16.02	15.93	13.08	0.330	<0.001	<0.001	0.764
Average daily gain (kg/day)	0.143	0.176	0.175	0.144	0.002	<0.001	<0.001	0.660
Feed-to-gain ^2^ (kg/kg)	7.34	7.21	6.90	7.66	0.242	0.624	0.002	0.930
Rumen Ph	6.49	6.43	6.43	6.49	0.031	0.053	0.109	0.977
Fecal Score	1.14	1.14	1.17	1.11	0.025	0.826	0.043	0.297

^1^ DMI, dry matter intake; ^2^ Feed: gain calculated as kg of dry matter consumed/kg of weight gain; ^3^ SEM = standard error mean, ^4^ PF = physical form (conventional vs. pelleted); SL = straw level (WS15 = wheat straw 15%, WS25 = wheat straw 25%), PF × SL = physical form × straw level of diets.

**Table 4 animals-10-01861-t004:** Body condition score (BCS) and structural measurements in fattening goats fed conventional and pelleted total mixed ration.

Measurement	Treatments				SEM ^2^	*p*–Value ^3^		
	PF		SL					
	Conventional	Pelleted	WS15	WS25		PF	SL	PF × SL
Gain in BCS ^1^	0.68	0.93	1.00	0.62	0.256	0.338	0.155	0.338
Gain in body length (cm)	4.62	4.87	4.89	4.60	0.464	0.594	0.540	0.936
Gain in heart girth (cm)	3.15	3.19	3.47	2.86	0.354	0.902	0.099	0.930
Gain in wither height (cm)	2.55	2.59	2.66	2.48	0.320	0.907	0.563	0.938
Gain in hip height (cm)	2.68	2.81	2.78	2.71	0.241	0.600	0.788	0.973

^1^ BCS = body condition score, performed at 0, 30, 60, and 90 days of the experiment using a 1–5 scoring system described by [15]. ^2^ SEM = standard error mean, ^3^ PF = physical form (conventional vs. pelleted); SL = straw level (WS15 = wheat straw 15%, WS25 = wheat straw 25%), PF × SL = physical form × straw level of diets.

**Table 5 animals-10-01861-t005:** Effects of conventional and pelleted TMR on nutrient digestibility coefficients in fattening goats.

Digestibility (%) ^1^	Treatments				SEM ^2^	*p*–Value ^3^		
	PF		SL					
	Conventional	Pellet	WS15	WS25		PF	SL	PF × SL
DM	70.32	68.90	69.94	69.28	2.31	0.549	0.780	0.813
OM	73.61	73.43	73.09	73.95	2.14	0.936	0.692	0.947
CP	61.44	65.03	64.17	62.30	3.30	0.294	0.579	0.842
aNDF	52.68	51.22	52.97	50.93	2.94	0.625	0.499	0.914
ADF	52.53	50.67	51.89	51.31	3.72	0.625	0.878	0.726
EE	79.82	79.10	79.58	79.34	2.48	0.776	0.926	0.769

^1^ DM, dry matter; OM, organic matter; aNDF: neutral detergent fiber inclusive of residual ash, ADF: acid detergent fiber inclusive of residual ash; EE, ether extract. ^2^ SEM = standard error mean, ^3^ PF = physical form (conventional vs. pelleted); SL = straw level (WS15 = wheat straw 15%, WS25 = wheat straw 25%), PF × SL = physical form × straw level of diets.

**Table 6 animals-10-01861-t006:** Effects of conventional and pelleted TMR on nitrogen balancing in fattening goats.

Parameters	Treatments				SEM ^1^	*p*–Value ^2^		
	PF		SL					
	Conventional	Pellet	WS15	WS25		PF	SL	PF × SL
N intake (g/day)	24.02	26.17	24.83	25.36	2.35	0.374	0.825	0.754
Fecal N (g/day)	4.87	4.61	4.72	4.76	0.569	0.651	0.953	0.777
Urinary N (g/day)	4.32	4.51	4.14	4.80	0.787	0.915	0.415	0.795
Retained N (g/day)	14.71	17.04	15.95	15.79	1.71	0.193	0.925	0.828

^1^ SEM = standard error mean, ^2^ PF = physical form (conventional vs. pelleted); SL = straw level (WS15 = wheat straw 15%, WS25 = wheat straw 25%), PF × SL = physical form × straw level of diets.

**Table 7 animals-10-01861-t007:** Effects of experimental diets on serum enzymes and on bilirubin and blood metabolites in fattening goats fed conventional and pelleted total mixed rations with two levels of wheat straw.

Item ^1^	Treatments				SEM ^2^	*p*–Value ^3^		
	PF		SL					
	Conventional	Pellet	WS15	WS25		PF	SL	PF × SL
ALP (IU/L)	161.82	161.54	162.60	160.76	4.210	0.946	0.662	0.501
ALT (IU/L)	22.32	22.97	22.86	22.43	0.888	0.463	0.630	0.083
AST (IU/L)	47.47	46.81	47.02	47.26	1.57	0.680	0.879	0.441
Bilirubin (mg/dL)	0.56	0.56	0.56	0.57	0.019	0.882	0.849	0.584
Glucose (mg/dL)	80.97	81.64	81.20	81.40	0.626	0.288	0.750	0.613
BUN (mg/dL)	19.50	19.45	19.28	19.67	0.292	0.867	0.178	0.593
Cholesterol (mg/dL)	42.91	43.90	43.37	43.44	0.290	0.175	0.928	0.442

^1^ ALP, alkaline phosphatase; ALT, alanine transaminase; AST, aspartate transaminase; BUN, blood urea nitrogen. ^2^ SEM = standard error mean, ^3^ PF = physical form (conventional vs. pelleted); SL = straw level (WS15 = wheat straw 15%, WS25 = wheat straw 25%), PF × SL = physical form × straw level of diets.

**Table 8 animals-10-01861-t008:** Complete blood count components of fattening goats fed conventional and pelleted total mixed rations with two different levels of wheat straw.

Blood Components ^1^	Reference Values ^2^	Treatments				SEM	*p*–Value ^3^		
		PF		SL					
		Conventional	Pellet	WS15	WS25		PF	SL	PF × SL
RBCs 10^12^/L	8–18	14.80	15.78	14.61	15.96	0.841	0.247	0.112	0.149
WBCs 10^9^/L	4–13	11.69	11.63	11.16	12.17	0.834	0.944	0.234	0.209
Hemoglobin g/dL	8–12	9.06	9.15	9.12	9.09	0.304	0.767	0.919	0.893
Lymphocytes 10^9^/L	2–9	6.93	7.14	7.22	6.84	0.494	0.672	0.450	0.359
Monocytes 10^9^/L	0–0.6	0.51	0.51	0.52	0.51	0.038	0.940	0.910	0.978
MCV fL	16–25	15.76	17.10	16.53	16.34	0.638	0.060	0.771	0.306
MCHC g/L	300–600	311.8	306.2	304.6	313.4	7.533	0.455	0.237	0.660
Hct %	22–38	30.73	31.47	30.73	31.46	1.240	0.553	0.560	0.376

^1^ RBCs, red blood cells; WBCs, white blood cells; MCV, mean corpuscle volume; MCHC, mean corpuscular hemoglobin concentration; Hct; hematocrit; ^2^ Adapted from Radostits et al. [21]. SEM = standard error mean, ^3^ PF = physical form (conventional vs. pelleted); SL = straw level (WS15 = wheat straw 15%, WS25 = wheat straw 25%), PF × SL = physical form × straw level of diets.

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
