# Peer review of "Effect of Physical Form and Level of Wheat Straw Inclusion on Growth Performance and Blood Metabolites of Fattening Goat"

_animals, 2020, doi:10.3390/ani10101861_

Round 1
Reviewer 1 Report
The Authors revised the text and made most of the corrections that were requested. Even if some information were not available (chewing activity, DIP/UIP), the results and the discussion were supported by new data and calculations. In my opinion this paper can be published in its present form.
Author Response
Reviewer 1 is agreed with our inputs, so there is no further suggestions
Reviewer 2 Report
Dietary high crude fiber can prevent ruminal acidosis in fattening goats. The present study compared the responses of two different CF level diets on the fattening goats. The results will be helpful to use the SW in the practical production. Minor comments: The authors should specify the NDF, ADF contents in different treatments in abstracts. And your also should provide the information of the CP, NE these key nutrient levels in diets. In my opinion, one factor of this trail is the fiber level. The CTMR15 (containing ?? NDF, ?? ADF mainly from 15% SW and 13% Soy hulls). The same to the other treatment. The description of present study makes the reader confusion, not understand the differences in the diets. In results description, the authors should provide the detailed information, i.e. AST, ALP, In conclusion, the authors can conclude the results. Please did not magnify the findings in this study. Only 15% and 25 SW two levels occur in this paper, how to give the conclusion “without noticeable adverse effects”. Please added “as compared to the 25% WS treatments. Please merge table 1 and 2 into one table. In tables, p-value of 0.000 should be changed intoAuthor Response
Please see the attachment

This manuscript is a resubmission of an earlier submission. The following is a list of the peer review reports and author responses from that submission.
Round 1
Reviewer 1 Report
In this paper, the effects of pelleted and conventional TMR with different amounts of wheat straw fed to fattening bucks are investigated. According to conclusions, there is the need for further studies to delineate the effect of particle size of straw and fiber content of the diet on growth performance of animals.
Introduction
The use of wheat straw (WS) for sheep and goat nutrition is widely studied; the influence of the physical form of WS to the dry matter intake (DMI) and to the performances of the animals are well known. The introduction does not report these information, that are necessary to understand the objectives of the trial. Please clearly describe why 2 levels of WS were chosen.
Materials and Methods
The physically effective NDF (peNDF) of diets and the chewing activity have not been investigated; these information are essential to understand the role of particle size on rumen DMI, rumen pH, etc.
Explain the meaning of acronyms (WS and PF)
This chapter is confused and it is difficult to read it; reorganize and rewrite the whole chapter; clearly separate the methods for measuring the growth performances of animals fed different diets from the digestibility trial and the analysis of metabolic parameters (follow the structure of chapter "Results").
The fecal score analysis is in sub-chapter 2.1 "Experimental Design and Animal Husbandry"); modify
Table 1: split into 2 tables (ingredients and chemical composition)
Digestibility coefficients of nutrients: describe calculation. Add CP digestibility coefficient
Feces and urine collection: describe the sampling methods
Nitrogen balance: describe calculation of retained N (g/day and %). Total protein of diets were similar among diets; rumen degradable (RDP or DIP) and rumen soluble protein of diets should be determined.
Blood collection: list all the performed analyses of blood and serum ("Results" report data from many parameters not listed in "Materials and Methods", i.e. hepatic enzymes). Explain the meaning of the analyses parameters.
Results
In all tables: do not repeat the meaning of acronyms
Explain what the "feed-to-gain ratio" is, and how it is calculated (in Table 2 is reported as "Feeds efficiency, kg/kg")
Discussion
Line 239-242: move to introduction
High NDF does not necessary mean low risk of rumen acidosis (33,7 vs 38,7% in this trial). Physically effective NDF (peNDF) of TMRs and chewing activity must be considered.
The discussion is mainly based on differences between animals affected, or not, by sub-acute rumen acidosis (SARA), and the well known role of fiber (NDF, peNDF) and saliva on preventing the SARA. As expected, animals of this trial did not experienced SARA; the Authors should orient the discussion towards the efficiency of the different particle size of straw or TMRs on animal performances, as stated in "Conclusions". Therefore, data on peNDF of straw and TMRs are needed for a better comprehension and discussion of the Results.
Reviewer 2 Report
The results of this study seem to find that feeding pelleted TMR with 15% WS improved DMI and growth performance. However, there are two key points the authors must reconsider. 1. I did not agree with the experimental design. For the different percent WS diet, the authors changed other ingredients, therefore, the action of this diet was not only due to the difference in the WS percent. In my opinion, the diets without WS should use the same ratios of different ingredients. 2. SE of initial BW of bulk is the 0.59. In my opinion, the authors should use the initial BW of bulk as the random effect. For all tables, the interaction between PF and SL did not significantly affect the parameters. Therefore, the authors should show the readers the main effects of PF and SL, not the interaction. Therefore, I suggest the authors to reconstruct the tables, and describe the suitable results. The present design included the effect of physical form, different diet, not only the SW, and their interaction.